# Omni-Angle Assault: An Invisible and Powerful Physical Adversarial Attack on Face Recognition

Shuai Yuan [1]  Hongwei Li [1]  Rui Zhang [1]  Hangcheng Cao [2]
Wenbo Jiang [1]  Tao Ni [2]  Wenshu Fan [1]  Qingchuan Zhao [2]  Guowen Xu [1]

## Abstract

Deep learning models employed in face recognition (FR) systems have been shown to be vulnerable to physical adversarial attacks through various modalities, including patches, projections, and infrared radiation. However, existing adversarial examples targeting FR systems often suffer from issues such as conspicuousness, limited effectiveness, and insufficient robustness. To address these challenges, we propose a novel approach for adversarial face generation, UVHat, which utilizes ultraviolet (UV) emitters mounted on a hat to enable invisible and potent attacks in black-box settings. Specifically, UVHat simulates UV light sources via video interpolation and models the positions of these light sources on a curved surface, specifically the human head in our study. To optimize attack performance, UVHat integrates a reinforcement learning-based optimization strategy, which explores a vast parameter search space, encompassing factors such as shooting distance, power, and wavelength. Extensive experimental evaluations validate that UVHat substantially improves the attack success rate in black-box settings, enabling adversarial attacks from multiple angles with enhanced robustness.

## 1. Introduction

Recently, face recognition (FR) systems utilizing deep learning models have seen widespread adoption across various domains, including financial transactions (Aru & Gozie, 2013; Bodepudi & Reddy, 2020), airport security (Rajamäki

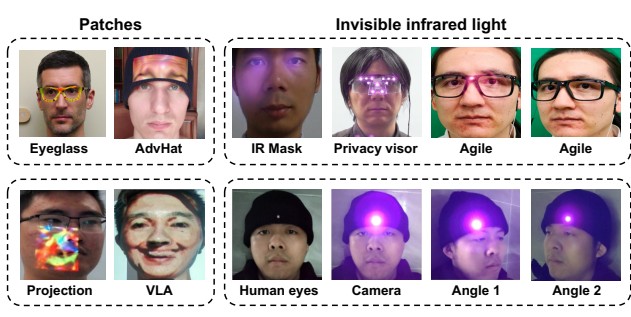

Figure 1: Physical adversarial attack on FR systems. (1) Patches attached to glasses or hats. (2) Visible projections cast onto the face. (3) Invisible infrared light on hats or glasses. (4) Our invisible and powerful UVHat, which remains effective across multiple angles.

et al., 2009; Zhu & Wang, 2020), access control (Ibrahim & Zin, 2011; Lee et al., 2020), crime prevention (Purshouse & Campbell, 2019; Archana et al., 2024), and law enforcement (Sajjad et al., 2020; Lynch, 2020), primarily due to their high accuracy in identifying individuals. These systems offer substantial benefits in enhancing security, efficiency, and convenience. However, recent studies (Goswami et al., 2018; Dong et al., 2019; Zhong & Deng, 2020) have revealed that FR systems are vulnerable to adversarial examples, carefully crafted inputs designed to mislead the model into making incorrect predictions. Such errors can lead to severe consequences, such as unauthorized access to restricted areas, wrongful identification, and even false arrests. Additionally, adversarial manipulations could compromise individual privacy, enable identity theft, or damage reputations. These vulnerabilities raise critical concerns about the reliability and security of FR systems, particularly in high-stakes applications where attackers may exploit these weaknesses to cause harm.

However, current adversarial attack methods exhibit significant limitations. Early research predominantly focused on adversarial examples in the digital domain (Garofalo et al., 2018; Kwon et al., 2019; Goodman et al., 2020), where attackers manipulated facial inputs at the pixel level. These digital approaches, while conceptually promising, are neither practical nor sufficiently effective for real-world

---

[1]School of Computer Science and Engineering, University of Electronic Science and Technology of China, Chengdu, China [2]Department of Computer Science, City University of Hong Kong, Hong Kong, China. Correspondence to: Guowen Xu <guowen.xu@uestc.edu.cn>.

*Proceedings of the 42nd International Conference on Machine Learning*, Vancouver, Canada. PMLR 267, 2025. Copyright 2025 by the author(s).

applications. In response to these issues, considerable efforts have been directed toward the development of physical adversarial examples that are capable of withstanding real-world environmental conditions. Physical adversarial perturbations targeting FR systems can be broadly categorized into patches, visible light, and invisible light-based attacks. Specifically, patches refer to adversarial perturbations that are printed by the attacker and subsequently attached to objects such as eyeglass frames (Sharif et al., 2016), hats (Komkov & Petiushko, 2021), masks (Zolfi et al., 2022), or even directly onto the face (Yin et al., 2021; Wei et al., 2022a). In contrast, visible light attacks typically involve the projection of adversarial perturbations (Nguyen et al., 2020) onto the attacker's face. Despite their potential, adversarial examples generated by these two attack types are hindered by substantial limitations in stealth and often appear visually conspicuous. To mitigate these shortcomings, recent studies (Yamada et al., 2013; Wang et al., 2024) have investigated the use of invisible infrared (IR) light to create adversarial perturbations, thereby enhancing the stealthiness of the attack. However, although this approach offers improvements in terms of concealment, the relatively weak energy associated with the IR light wavelength (Vaia, 2024) results in diminished effectiveness. Furthermore, as demonstrated in Figure 1, the deployment of IR emitters on eyeglasses (e.g., the Agile method) fails to support effective attacks from multiple angles, leading to reduced robustness. Moreover, directing IR light at the face (Zhou et al., 2018) introduces significant safety concerns, as it may pose risks to the attacker's vision, thereby further limiting its practical applicability.

To overcome the aforementioned limitations, we introduce a novel physical adversarial attack, UVHat, which utilizes invisible ultraviolet (UV) light emitted from a hat to disrupt FR models. In contrast to prior approaches, the primary challenges in developing UVHat lie in accurately simulating UV light sources on a curved surface and determining the optimal attack parameters in a black-box setting. Our method can be conceptualized as a three-step process. First, we devise an interpolation-based UV simulation technique that leverages a video interpolation model to generate UV images under varying distances, powers, and wavelengths within the digital domain. Second, we introduce a hemispherical UV modeling strategy to update the relevant parameters based on the positions across the curved surface. Finally, we employ a reinforcement learning optimization approach, wherein the agent iteratively explores the parameter space to identify the most effective attack parameters.

We conduct experiments in the physical world using two datasets and four models, comparing our approach with two baseline methods. Ablation studies analyze the impact of factors such as wavelength and power on attack performance. Additionally, we test the robustness of our method

under real-world conditions, including angle and lighting intensity. Finally, we discuss potential defense strategies and limitations, providing insights into protecting FR systems from UVHat.

Our contributions are summarized as follows:

- We propose a novel physical adversarial attack using UV light, i.e., UVHat, which generates invisible and powerful adversarial perturbations to fool FR systems.

- We design an interpolation-based UV simulation and a hemispherical UV modeling method to simulate UV light sources on curved surfaces and utilize reinforcement learning to search for optimal attack parameters.

- Extensive experiments validate the effectiveness of UVHat against FR models in a black-box setting.

## 2. Background and Related Works

In this section, we provide an overview of FR systems and the evolution of physical adversarial examples targeting these systems.

### 2.1. Face Recognition Systems

Face recognition has always been one of the most popular research topics in the field of computer vision. In recent years, with the rapid development of deep learning technology, face recognition has achieved significant success in various applications, such as DeepFace (Taigman et al., 2014), FaceNet (Schroff et al., 2015) and ArcFace (Deng et al., 2019).

In general, the deep learning-based face recognition system can be mainly divided into four steps: face detection, face alignment, feature extraction, and feature matching. Firstly, the face detection step is to detect and locate facial regions in input images or videos. Representative face detection methods include MTCNN (Zhang et al., 2016), RetinaFace (Deng et al., 2020) and YOLO-Face (Chen et al., 2021), etc; Secondly, to improve the accuracy of subsequent feature extraction, detected faces usually need to be aligned. A common practice is to align the face to a standard template by detecting key points of the face (e.g., eyes, tip of the nose, corners of the mouth, etc.) and using geometric transformations. After that, the feature extraction step transforms raw face images into discriminative feature vectors for the subsequent identity verification or classification. State-of-the-art (SOTA) face feature extraction models includes Deep-Face (Taigman et al., 2014), FaceNet (Schroff et al., 2015), OpenFace (Baltrušaitis et al., 2016), ArcFace (Deng et al., 2019), AdaFace (Kim et al., 2022), TransFace (Dan et al., 2023), etc. Finally, the feature matching step compares the extracted feature vectors with known face features in the database to determine identity.

## 2.2. Physical Adversarial Example against FR

Physical adversarial attacks against FR systems have gained significant attention due to their practicality and real-world implications. Unlike digital adversarial attacks that manipulate pixel values directly, physical attacks introduce perturbations to physical objects, such as glasses, makeup, or masks, that are then captured by cameras.

One of the most prominent methods in physical adversarial attacks is the use of adversarial patches. For instance, Sharif et al. proposed the use of adversarially crafted eyeglass frames to bypass FR systems (Sharif et al., 2016); Komkov et al. introduced adversarial hats (Komkov & Petiushko, 2021), demonstrating that such patches could mislead FR systems under various lighting and environmental conditions; Lin et al. developed a framework that uses adversarial makeup to deceive FR systems, ensuring the patterns remain imperceptible to humans (Lin et al., 2022).

On the other hand, many studies have utilized visible light and invisible light to construct physical adversarial examples. For instance, Nguyen et al. used adversarial light projections to conduct real-time physical attacks on FR systems (Nguyen et al., 2020). Similarly, Zhou et al. deployed IR emitters mounted on hats to illuminate faces and bypass FR systems (Zhou et al., 2018). Additionally, Wang et al. designed adversarial glasses equipped with IR lasers to launch dodging and impersonation attacks against FR systems (Wang et al., 2024). Compared to works using patches and visible light, our UV light is invisible to the naked eye, offering superior stealth. This is because UV's shorter wavelengths (below 400nm) fall outside the visible spectrum (400nm-700nm). In contrast to methods employing IR light, our use of UV light provides higher energy, enabling stronger adversarial perturbations and achieving a higher attack success rate. Furthermore, unlike adversarial glasses with IR lasers (Yamada et al., 2013) (Wang et al., 2024), our method succeeds in attacks from multiple angles. Notably, methods that project IR light directly onto the face (Zhou et al., 2018) pose a risk of harming the attacker's eyes, a drawback that our approach completely eliminates. A detailed comparison can be found in Appendix A.

## 3. Threat Model

**Attacker's capabilities.** Similar to previous works (Dong et al., 2019; Wei et al., 2022b; Wang et al., 2024), we assume a black-box setting, where an attacker cannot directly access the internal information of the model, including its architecture, parameters, and gradients. Compared to the white-box setting, the black-box setting more accurately reflects real-world attack scenarios, such as financial monitoring and airport security.

**Attacker's goals.** We consider three types of attack goals:

(1) Dodging attacks: The attacker's face is *in* the database, but the FR system misidentifies the attacker as a different individual:

$$f(UVHat(x_a)) \neq y_a, s.t. y_a \in D_{identity} \qquad (1)$$

where $f(\cdot)$ represents FR model, $x_a$ denotes the attacker's face with identity $y_a$ (belongs to identity dataset $D_{identity}$). A practical example is an attacker on a blacklist being misclassified as someone off the list, thereby gaining unauthorized access to restricted areas such as government buildings.

(2) Denial-of-Service (DoS) attacks: The attacker causes the FR system to fail to detect any face, resulting in no identification:

$$f(UVHat(x_a)) = None \qquad (2)$$

This type of attack prevents the FR system from recognizing anyone, potentially disrupting operations such as fugitive tracking.

(3) Impersonation attacks: The attacker's face is *not in* the database, but the FR system misidentifies the attacker as any identity (i.e., untargeted)in the database:

$$f(UVHat(x_a)) \neq y_a, s.t. y_a \notin D_{identity} \qquad (3)$$

where the attacker's identity $y_a$ is absent from the database, and misclassification into any identity in $D_{identity}$ is a success. We also consider a specific identity (i.e., targeted):

$$f(UVHat(x_a)) = y_{target}, s.t. y_a \notin D_{identity} \qquad (4)$$

Real-world examples include an outsider impersonating an employee to gain entry to a company (untargeted), or an attacker attempting to unlock a smartphone or access a secured facility by mimicking a specific authorized individual (targeted).

## 4. Methodology

In this section, we present the contributions of our methodology. We begin by proposing the interpolation-based UV simulation method to simulate UV light sources under varying distances, powers, and wavelengths. Unlike previous approaches that focus solely on flat surfaces, we design the hemispherical UV modeling approach to analyze the UV intensity at different positions on curved surfaces. Finally, in a black-box setting, we leverage reinforcement learning to identify the optimal attack parameter for FR systems, enabling effective deployment in the physical world.

### 4.1. Interpolation-based UV Simulation

The intensity of UV light in the real world is affected by various factors, such as distance, power, and wavelength.

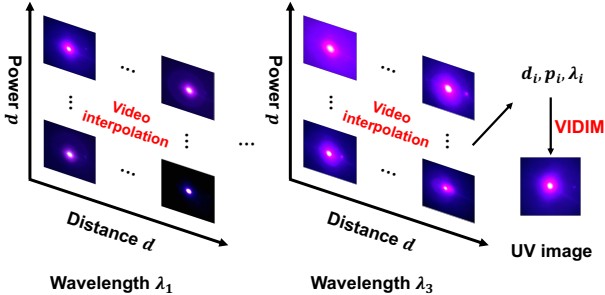

Figure 2: Simulating UV light with the VIDIM model.

Moreover, UV beams are diffuse, making it a significant challenge to simulate UV light in digital environments.

According to (Diffey, 2002), the intensity $I$ of UV light is given by:

$$I(d, p, \lambda) = \frac{p \cdot \lambda^2}{4\pi nd^2} \tag{5}$$

where $d$ is the distance from the light source to the receiving point, $p$ represents the power of the light source, $\lambda$ is the wavelength of UV light, and $n$ is the refractive index of air, typically taken as a constant value of 1.0003. For simplicity, environmental factors such as temperature and humidity are neglected in this model.

Inspired by (Reda et al., 2022), we do not simulate luminous images of UV light directly. Instead, we capture multiple images of UV light emission in the physical world, recording three parameters for each image: the distance $d$, the power $p$, and the wavelength $\lambda$. Since it is impractical to record UV images under all possible combinations of parameters, we employ a video interpolation model called VIDIM (Jain et al., 2024). VIDIM generates short videos given a start and end frame, utilizing cascaded diffusion models to produce all intermediate frames through joint denoising and fast sampling. As illustrated in Figure 2, we first group the UV light emissions by wavelength $\lambda_i$. For UV light with the same wavelength, we record their images at various distances and powers. Given any set of parameters, we first identify the corresponding group based on wavelength and then determine the start and end frames based on distance and power. Finally, the VIDIM model is used to generate the corresponding UV images.

### 4.2. Hemispherical UV Modeling

We have already established a method to determine the corresponding UV images based on power, wavelength, and distance. The next step is to place the simulated UV light source on a hat. Unlike the flat surfaces used by (Wang et al., 2024), our hat has a curved surface, and the placement of the UV emitter at different positions further affects the UV intensity. The UV light positioned at different locations on the curved surface cannot be simply rotated as in the case of a flat image. The shape of the UV light does not

change, but rather the intensity of the light is reduced due to the varying emission angles. Therefore, it is a challenge to simulate UV light sources at different locations on a curved surface.

Human head shapes vary, but the region covered by a hat can be approximated as a hemisphere. As shown in Figure 3, we model the hat as a hemisphere and assume that the camera is positioned in front of point A. Since the camera captures a 2D image, we can determine the radius of the hemisphere by calculating the difference in the Y-axis coordinates between points A and B, i.e., $r = (y_B - y_A)$. Next, we analyze how the UV light intensity changes at different positions on the curved surface.

Firstly, we assume that the UV light source is placed at point A, where the light emitter and the camera are aligned along the Z-axis. In this case, the UV intensity is not affected by the emission angle. Using the method described in Section 4.1, we can calculate the corresponding UV image based on parameters such as distance $d$, power $p$, and wavelength $\lambda$. Secondly, we observe point B, where the emission direction is along the Y-axis, perpendicular to the Z-axis. As a result, the light intensity captured by the camera is 0. Similarly, for point C, where the emission direction is along the X-axis, the light intensity is also 0. Thirdly, we analyze the UV intensity at point D. Points A, B, and D all lie in the Y-O-Z plane. As shown in Figure 3, from a side view, the emission angle at point D relative to the Z-axis is $\theta$. The angle $\theta$ can be calculated based on the positions of points A, B, and D:

$$\theta_D = arcsin(\frac{y_D - y_A}{r}) \tag{6}$$

Thus, the light intensity at point D is given by:

$$I_D = I(d, p, \lambda)cos(\theta_D) = \frac{p \cdot \lambda^2}{4\pi nd^2}cos(arcsin(\frac{y_D - y_A}{r})) \tag{7}$$

Assuming that the distance $d$ and wavelength $\lambda$ at point D remain unchanged, the power $p_D$ becomes:

$$p_D = \frac{I_D \cdot 4\pi nd^2}{\lambda^2} = \frac{I(d, p, \lambda)cos(arcsin(\frac{y_D - y_A}{r})) \cdot 4\pi nd^2}{\lambda^2} \tag{8}$$

Therefore, with the new power $p_D$, distance $d$, and wavelength $\lambda$, we can calculate the corresponding UV image and place it at point D. Similarly, as shown in Figure 3, the angle $\theta_E$ at point E can also be calculated:

$$\theta_E = arcsin(\frac{x_E - x_A}{r}) \tag{9}$$

Finally, we calculate the UV intensity at any arbitrary point F. Using the spherical equation $x^2 + y^2 + z^2 = r^2$, we can determine the 3D coordinates of point F, denoted as $(x_F, y_F, z_F)$. By projecting point F onto the Z-axis, we can compute the angle $\theta_F$ between point F and the Z-axis:

$$cos(\theta_F) = \frac{z}{r} = \frac{\sqrt{r^2 - x_F^2 - y_F^2}}{r} \tag{10}$$

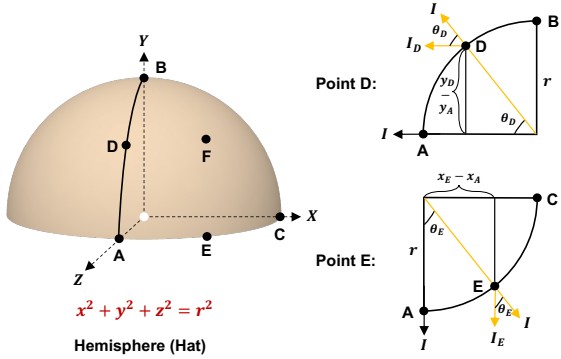

Figure 3: 3D hemisphere modeling with light intensity calculation at arbitrary positions.

Therefore, the power $p_F$ at point F is:

$$p_E = \frac{I(d,p,\lambda)\sqrt{r^2 - x_F^2 - y_F^2} \cdot 4\pi nd^2}{r\lambda^2} \quad (11)$$

By calculating the angle $\theta_F$, we can determine the UV light intensity at any point E on the hat and compute the corresponding power $p_E$. Using $p_E$, the distance $d$, and the wavelength $\lambda$, we can obtain the UV image at point E with the VIDIM model and add the image to point E.

### 4.3. Optimization

Although the aforementioned methods can simulate UV light sources on curved surfaces, simply optimizing attack parameters presents significant challenges. First, the black-box setting and the discrete nature of certain parameters make gradient-based optimization strategies unfeasible, while exhaustive search methods are computationally expensive. Second, placing the UV light source with maximum power at the closest distance may easily lead to DoS attacks, but it does not achieve other attack goals such as impersonation or dodging.

Based on these observations, we design a reinforcement learning approach to avoid the need for hand-crafted heuristics. We first define the search space and action space based on the methods outlined in Section 4.1 and 4.2, and design reward functions for the agent tailored to different attack goals. Our approach is inspired by the actor-critic framework (Gruslys et al., 2018), where the actor selects the optimal action through a policy network, while the critic evaluates the value of the action. Finally, we obtain the perturbation parameters, i.e., distance, power, wavelength, and position, to deploy the attack in the physical world.

Specifically, our process works as follows. First, we define the attack parameters $\phi = [d, p, \lambda, (x, y)]$ as the state space, where $d \in [d_{min}, d_{max}]$, $p \in [p_{min}, p_{max}]$, $\lambda \in \{365\,\text{nm}, 395\,\text{nm}, 415\,\text{nm}\}$, $x \in [0, X_{image}]$, $y \in [0, Y_{image}]$. Next, we define the agent's action space. At time $t$, the actor select an action $a_t = [d_t, p_t, \lambda_t, (x_t, y_t)]$

based on the current state $s_t$, according to the policy network $\pi$. The $\pi(a_t|s_t)$ represents the probability of selecting action $a_t$ given the state $s_t$. Based on the method described in Section 4.2, the corresponding attack parameters at position $(x_t, y_t)$ are determined as $a'_t = [d_t, p'_t, \lambda_t, (x_t, y_t)]$. Finally, using the VIDIM model, the corresponding UV image is generated based on these parameters.

Next, the agent performs action $a'_t$, which adds the UV light to the normal sample to obtain a reward. Since the attack goals are different, we define a combined reward function as follows:

$$R_{total} = k_1 R_{pos} + k_2 R_{im\_un} + k_3 R_{im\_tar}$$
$$+ k_4 R_{dodg} + k_5 R_{DoS}; \sum_{i=1}^{5} k_i = 1 \quad (12)$$

Different models compute euclidean distance or cosine similarity for embedding features. To improve clarity, we express FR results as probability $p$ and explain how the probability $p$ is derived from euclidean distance or cosine similarity in Appendix B. We introduce each reward function separately. First, $R_{pos}$ determines whether the position of the UV light source is out of range of the hat:

$$R_{pos} = -\rho \cdot mask(x_t, y_t) \quad (13)$$

where $\rho$ is a penalty coefficient and $mask$ is a function that returns 0 when the position of the UV light source $(x_t, y_t)$ is within the region of the hat, and 1 otherwise. Second, for untargeted impersonation attacks, the reward function is:

$$R_{im\_un} = max(p_1, ..., p_N) \quad (14)$$

where $p_i$ is the probability of the $i$-th category and N is the number of categories. The goal of the attack is to maximize the model's classification probability, without focusing on any specific identity. For targeted impersonation attacks, the reward function is:

$$R_{im\_tar} = \eta \cdot p_{target} \quad (15)$$

where $\eta$ is a reward coefficient and $p_{target}$ represents the probability that the model classifies the attacker into the target category. Third, for dodging attacks, the reward function is:

$$R_{dodg} = -\tau \cdot p_{original} \quad (16)$$

where $\tau$ is a penalty coefficient. Note that, at this point, the attacker belongs to the database, and the corresponding probability is $p_{original}$. Therefore, the attacker aims to minimize the probability of the original identity as much as possible. Finally, for DoS attacks, the reward function is:

$$R_{DoS} = \sum_{i=1}^{N} p_i \cdot log(p_i) \quad (17)$$

**Algorithm 1** The actor-critic pseudocode
___
Initialize the parameters of actor network $\theta$ and critic network $\phi$, learning rate $\alpha$ and $\beta$, $T$=0.
**repeat**
    Reset the gradients of actor and critic networks to 0.
    $t_{start} = 1, t = 1$.
    Get state $s_t$.
    **repeat**
        Perform $a'_t$ according to policy $\pi_\theta(a_t|s_t)$.
        Receive reward $R_t$ and new state $s_{t+1}$.
        Compute TD error to update critic network:
        $V_\phi(s_t) \leftarrow V_\phi(s_t) + \alpha(R_{t+1} + \gamma V_\phi(s_{t+1}) - V_\phi(s_t))$.
        Calculate the advantage function:
        $A_t = Q(s_t, a'_t) - V_\phi(s_t)$.
        Update the actor network:
        $\theta_{t+1} = \theta_t + \beta \nabla_\theta log \pi_\theta(a'_t, s_t) A(s_t, a'_t)$.
        $t \leftarrow t + 1$.
    **until** $t - t_{start} > t_{max}$
    $T \leftarrow T + 1$
**until** $T > T_{max}$
___

We encourage the model to output a uniform distribution by maximizing entropy, which represents the level of uncertainty in the distribution, thereby achieving DoS attacks.

After obtaining the reward, we first introduce the state-valued function $V$ and the action-valued function $Q$ used by the critic network:

$$V(s_t) = \mathbb{E}\left[\sum_{t=0}^{\infty} \gamma^t R_t | s_0 = s_t\right] \quad (18)$$

$$Q(s_t, a'_t) = \mathbb{E}\left[\sum_{t=0}^{\infty} \gamma^t R_t | s_0 = s_t, a_0 = a'_t\right] \quad (19)$$

The function $V$ represents the expected return when following the current policy from state $s_t$, while the function $Q$ is the expected cumulative reward after taking action $a'_t$ in state $s_t$. The critic updates the value functions through the Temporal Difference (TD) error:

$$V(s_t) \leftarrow V(s_t) + \alpha(R_{t+1} + \gamma V(s_{t+1}) - V(s_t)) \quad (20)$$

where $\alpha$ is the learning rate, and $\gamma$ is the discount factor. The core idea of TD error is to improve the prediction of the current state value by using the current estimate and the estimated value of the next state, rather than waiting until the end of the entire episode to perform an update.

To measure how much additional reward the action $a'_t$ brings, we compute the advantage function $A_t$:

$$A(s_t, a'_t) = Q(s_t, a'_t) - V(s_t) \quad (21)$$

The advantage function helps in updating the parameter of the actor:

$$\theta_{t+1} = \theta_t + \beta \nabla_\theta log \pi_\theta(a_t, s_t) A(s_t, a'_t) \quad (22)$$

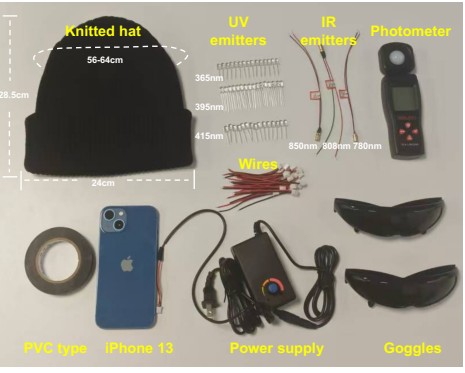

Figure 4: Experimental hardware used in the real-world environments.

where $\beta$ is the learning rate. Thus, the critic assists the actor in improving its policy, enabling it to select the optimal actions. Ultimately, the agent identifies the best attack parameters for deployment in the physical world. The entire process is summarized in Algorithm 1.

## 5. Physical Evaluation

Here, we present our physical experimental setup to evaluate the performance of UVHat in attacking four models across two datasets, comparing it with two baselines. Additionally, we provide key results from ablation studies and assess the robustness of the attack in real-world environments.

### 5.1. Experimental Setup

**Hardware.** Figure 4 illustrates the various devices used in our experiments. The UV emitters have three wavelengths of $365\,\text{nm}$, $395\,\text{nm}$, and $415\,\text{nm}$, while the IR emitters have three wavelengths of $780\,\text{nm}$, $808\,\text{nm}$, and $850\,\text{nm}$. We connect the emitters to the power supply using connecting wires and PVC tape. The power supply is an adjustable voltage unit designed specifically for lasers, with a voltage range of 1-5V. UVHat incorporates UV emitters into a knitted hat, as shown in Figure 4. We use an iPhone 13 equipped with the Sony IMX772 CMOS to capture facial images. To ensure experimental safety, we equip ourselves with goggles that can filter light wavelengths ranging from $200\,\text{nm}$ to $2000\,\text{nm}$. Additionally, we use a photometer to assess the ambient light intensity.

**Datasets.** We utilize two public datasets in the following experiments, i.e., LFW and CelebA. LFW (Huang et al., 2008) contains 13,233 images of 5,749 people and 1,680 of the people pictured have two or more distinct photos. CelebA (Liu et al., 2018) is a large-scale face attributes dataset with more than 200K images.

**Models.** We select four widely used face recognition models as target models, i.e., ArcFace (Deng et al., 2019), FaceNet (Schroff et al., 2015), CosFace (Wang et al., 2018), and Mo-

Table 1: The ASR of UVHat on various models in the physical world.

| Goal | Method | LFW | | | | CelebA | | | |
|---|---|---|---|---|---|---|---|---|---|
| | | ArcFace | FaceNet | CosFace | MobileFace | ArcFace | FaceNet | CosFace | MobileFace |
| DoS | MaxUV | 52% | 66% | 41% | 61% | 39% | 55% | 28% | 46% |
| | 2DUV | 4% | 12% | 9% | 6% | 5% | 10% | 2% | 6% |
| | UVHat | 72% | 89% | 80% | 78% | 67% | 88% | 70% | 73% |
| Dodging | MaxUV | 25% | 32% | 19% | 21% | 19% | 26% | 17% | 20% |
| | 2DUV | 2% | 11% | 0% | 4% | 4% | 7% | 1% | 2% |
| | UVHat | 81% | 100% | 78% | 87% | 75% | 91% | 64% | 83% |
| Untargeted Impersonation | MaxUV | 33% | 41% | 36% | 28% | 26% | 33% | 23% | 30% |
| | 2DUV | 4% | 13% | 5% | 9% | 1% | 2% | 0% | 2% |
| | UVHat | 77% | 93% | 84% | 80% | 75% | 86% | 81% | 80% |
| Targeted Impersonation | MaxUV | 3% | 2% | 0% | 4% | 0% | 0% | 0% | 2% |
| | 2DUV | 1% | 3% | 0% | 0% | 0% | 1% | 0% | 0% |
| | UVHat | 46% | 69% | 44% | 55% | 39% | 57% | 42% | 47% |

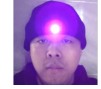 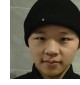 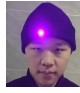 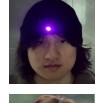 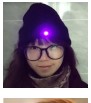 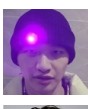

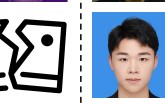 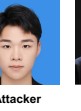 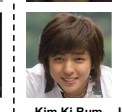 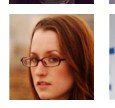 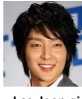

**DoS attacks**   **Dodging attacks**   **Impersonation attacks**

Attacker   Lupe Fiasco   Kim Ki Bum   Ingrid Michaelson   Lee Joon-gi

Figure 5: Adversarial attack examples for different goals. The first row shows the model inputs, while the second row displays images corresponding to the model's highest-confidence class.

bileFace (Chinaev et al., 2018), to comprehensively evaluate the effectiveness of our UVHat.

**Metric.** The attack success rate (ASR) is used as the evaluation metric, which is the percentage of successfully attacked samples among all the attacked samples. We set the decision threshold as 0.01 false acceptance rate for each victim model.

**Baselines.** We established two baselines for comparison. The first approach, denoted as **MaxUV**, places the highest-power UV emitter at the closest position to the camera. The second approach, referred to as **2DUV**, positions the UV emitter randomly, without considering the curved surface.

### 5.2. Overall Performance

We evaluate the performance of UVHat in the physical world and compare it with two baseline methods.

**Attack results of UVHat.** To verify the feasibility of UVHat, we train multiple models for different attack goals. For DoS and dodging attacks, we augment the pre-trained models with face images from five volunteers (specifically, 50 images per volunteer), and continue training until the model achieves a classification accuracy of over 95% for these individuals. For impersonation attacks, we directly utilize the pre-trained models. The targeted impersonation attack is considered successful if the attacker is classified as one of 10 randomly selected identities. We implement UVHat in real-world environments, and Figure 5 presents ex-

amples for various attack objectives. A camera can capture UV light because its wavelength falls within the camera's range but outside the visible spectrum.

We test the ASR of UVHat in real-world environments across multiple models, with the results presented in Table 1. The dodging attacks consistently achieve the highest ASR, with values reaching over 91% in the FaceNet model, while other results fall within the range of 64∼87%. The ASRs of DoS attacks are slightly lower than those of the dodging attack, with a maximum ASR reaching 89%. It is worth noting that increasing the UV emitter power can potentially lead to a higher ASR, as it may result in image overexposure. For impersonation attacks, the untargeted version outperforms the targeted version, as an attacker has a larger pool of identities to impersonate. The performance of the targeted impersonation attack is limited by the distance between the embedded classes, which can prevent successful attacks if the target's embedding is far from the adversarial example. Additionally, among the four models tested, FaceNet is the most vulnerable to these attacks, consistently exhibiting the highest ASR. In contrast, ArcFace and CosFace, both based on the ResNet architecture, show similar resistance to adversarial attacks.

**Comparison with two baselines.** We compare UVHat with two baselines, and the experimental results are shown in Table 1. The results indicate that MaxUV achieves the highest ASR of 66% only for DoS attacks. In all other cases, the attack success rate of MaxUV is below 41%. This aligns with our observations in the UVHat experiments, where the MaxUV favors DoS attacks by increasing the UV light coverage, which could potentially even damage the camera.

The performance of 2DUV is even worse, with none of the attack success rates exceeding 13%. This is because 2DUV lacks any optimization algorithm and does not consider surface curvature. As a result, the simulation results in the digital domain do not translate effectively into the physical world. For instance, a UV emitter using 4V in the simulation may only produce an effect equivalent to 3V when deployed in the physical world due to the impact of surface curvature. This discrepancy between the simulated and physical

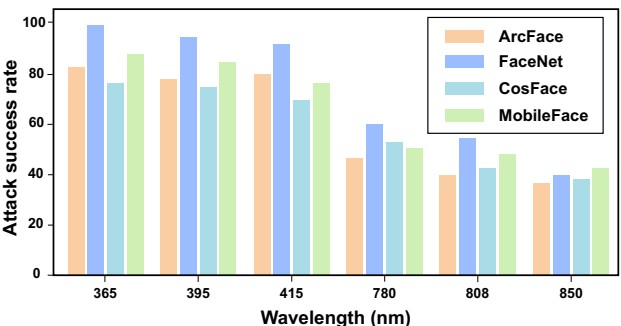

Figure 6: The ASR of UVHat at different wavelengths for dodging attacks.

Table 2: The ASR of UVHat at different voltages on Arc-Face trained in LFW.

| Goal | Voltage | | | | | | |
|---|---|---|---|---|---|---|---|
| | 1.0V | 1.5V | 2.0V | 2.5V | 3.0V | 4.0V | 5.0V |
| Dodging | 31% | 40% | 56% | 62% | 79% | 78% | 82% |
| DoS | 17% | 33% | 40% | 45% | 56% | 61 | 73% |
| $Imper_{un}$ | 23% | 38% | 52% | 68% | 75% | 75% | 76% |
| $Imper_{tar}$ | 10% | 25% | 36% | 40% | 44% | 31% | 23% |

world results leads to attack failure. Therefore, compared to both MaxUV and 2DUV, UVHat effectively simulates the UV light source on the hat in the physical world and optimizes the attack parameters. The real-world evaluation demonstrates its success in bypassing the FR systems.

### 5.3. Ablation Studies

In this section, we examine several possible factors that affect UVHat, including the wavelength, the UV power, and the number of UV emitters.

**Impact of the wavelength.** In this experiment, we fix the distance at 30 cm and set the voltage to 4V to evaluate the ASR of the dodging attack on four models trained on the LFW dataset. Note that, we add three IR emitters with varying wavelengths, i.e., 780 nm, 808 nm, 850 nm. As shown in Figure 6, the ASR gradually decreases as the wavelength increases. This occurs because longer wavelengths correspond to weaker energy, resulting in a reduction in the adversarial perturbations. However, an increase in wavelength enhances the light's penetration ability. As the distance increases, UV light is more easily absorbed and dissipates in the atmosphere, while IR light remains visible to the camera. For FR systems, which typically occur at close distances, stronger energy is required rather than enhanced penetration. Therefore, UV light is well-suited for attacking FR systems in such close-range scenarios.

**Impact of the UV power.** Here, we evaluate the performance of UVHat at different voltage levels, with a UV wavelength of 365 nm and a distance of 30 cm. Table 2 presents the ASR of UVHat for various attack goals on the

Table 3: The ASR of UVHat with varying numbers of UV emitters. "Num" represents the number of UV emitters. The result is presented as "A|B", where A represents untargeted impersonation attacks and B denotes DoS attacks.

| Num | ArcFace | FaceNet | CosFace | MobileFace |
|---|---|---|---|---|
| 1 | 75%\|71% | 90%\|87% | 80%\|76% | 80%\|76% |
| 2 | 80%\|69% | 91%\|89% | 82%\|81% | 79%\|81% |
| 3 | 82%\|75% | 93%\|90% | 85%\|85% | 81%\|85% |
| 4 | 80%\|76% | 87%\|93% | 80%\|85% | 82%\|85% |
| 5 | 81%\|83% | 86%\|94% | 77%\|87% | 78%\|87% |

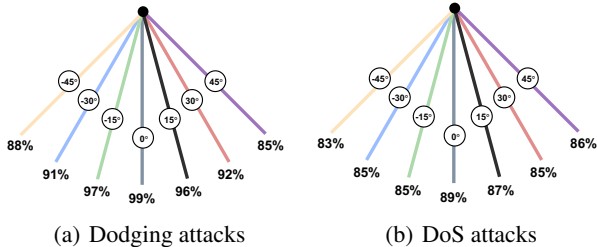

(a) Dodging attacks      (b) DoS attacks

Figure 7: The ASR of UVHat on FaceNet at different angles.

ArcFace model. As the voltage increases, the ASRs for different attacks improve to varying degrees. Notably, the DoS attack achieves the most significant increase, as higher voltage amplifies the adversarial perturbations, making DoS attacks more successful. It is important to note that the ASR for targeted impersonation attacks starts to decrease once the voltage exceeds 3V. This may be because the attacker can successfully classify the input into the target class with smaller perturbations, but larger perturbations prevent the input from crossing into the target class region.

**Impact of the number of UV emitters.** We expand the state space in Section 4.3 to $\phi = \{[d, p_i, \lambda_i, (x_i, y_i)]\}_{i=1}^C$, where $C$ is the number of UV emitters, allowing for the simultaneous optimization of multiple UV emitters. Note that each emitter has the same distance, but the wavelength and power can vary. Due to space limitations, we prioritize testing untargeted impersonation attacks and DoS attacks.

As shown in Table 3, the number of UV emitters has varying effects on different attack goals. For DoS attacks, increasing the number of emitters introduces more adversarial perturbations, thereby increasing the ASR. For untargeted impersonation attacks, the ASR increases most significantly when the number of emitters goes from 1 to 2. However, when the number exceeds 3, the ASR improvement becomes limited. In addition, adding more emitters may trigger a DoS attack, causing the untargeted impersonation attack to fail.

### 5.4. Robustness

To verify the robustness of UVHat in the physical world, we evaluate the ASR of UVHat under different environmental conditions, including angle, distance, and ambient light.

Table 4: The ASR of UVHat at varying distances for DoS attacks.

| Model | 25cm | 30cm | 35cm | 40cm | 45cm | 50cm |
|---|---|---|---|---|---|---|
| ArcFace | 78% | 73% | 72% | 70% | 67% | 54% |
| FaceNet | 91% | 92% | 89% | 80% | 74% | 65% |
| CosFace | 83% | 80% | 78% | 79% | 69% | 59% |
| MobileFace | 79% | 80% | 79% | 73% | 66% | 58% |

Table 5: The ASR of UVHat under different ambient light conditions for DoS attacks.

| Light | ArcFace | FaceNet | CosFace | MobileFace |
|---|---|---|---|---|
| 1000lux | 82% | 95% | 85% | 89% |
| 2000lux | 73% | 89% | 80% | 80% |
| 4000lux | 66% | 78% | 72% | 67% |
| 6000lux | 47% | 53% | 41% | 38% |

**Robustness to angles.** As shown in Figure 7, we test the ASR of dodging and DoS attacks at different angles, with different colored lines representing different angles. It is evident that the frontal view ($0°$ angle) typically achieves the highest ASR. Angle deviation only slightly reduces the ASR, with dodging attacks causing a maximum decrease of 14%, while DoS attacks experience a maximum decrease of 6%. This result further demonstrates the robustness of UVHat across multiple angles. Additionally, compared to DoS attacks, the ASR of dodging attacks is more sensitive to angle changes. This is likely because angular deviations affect facial features, making recognition more challenging.

**Robustness to distances.** To evaluate the impact of the distance between the person and the camera on UVHat's performance, we conduct DoS attacks while keeping other attack parameters constant. The results shown in Table 4 indicate that as the distance increases, the ASR gradually decreases. Specifically, when the distance exceeds $45\,\mathrm{cm}$, there is a sharp decline in ASR. This is due to the nature of UV light, which disperses outward and is easily absorbed by the atmosphere. As the distance increases, the adversarial perturbation captured by the camera becomes weaker. We would like to emphasize that the inability to successfully attack at long distances is acceptable, as FR systems typically operate within close-range distances.

**Robustness to ambient light.** We use a photometer to measure ambient light intensity and evaluate the ASR of the dodging attack under different lighting conditions. As shown in Table 5, the ASR decreases gradually with the increase in ambient light intensity. However, UVHat still maintains a relatively high ASR for FaceNet. It is important to note that all physical adversarial attack methods utilizing light are influenced by ambient lighting. Typically, FR systems operate indoors, where direct sunlight is avoided, resulting in lower ambient light intensity, which in turn benefits the effectiveness of UVHat attacks.

# 6. Discussion

## 6.1. Defenses.

We discuss three potential defense methods.

(1) **UV filter**: Installing a UV filter inside the camera can prevent the capture of UV light. However, attackers could bypass this defense by using wavelengths close to UV light.

(2) **Image detection**: This method involves detecting the presence of UV features in an image, such as color and shape. Since UV light naturally exists in daily life, and human hair, clothing, and circular decorations may resemble UV characteristics, such features could further interfere with the detection process.

(3) **Model robustness**: The most representative approach in this category is adversarial training, which is typically used to defend against specific attacks. However, UVHat involves numerous parameters, and adversarial training cannot encompass all variations of UV signals, leading to inadequate defense. Additionally, the need to generate a large number of adversarial images and retrain the FR model makes this approach costly.

## 6.2. Limitations.

Our approach performs poorly in extreme lighting conditions, which is a limitation common to all adversarial attacks relying on light signals. Additionally, UVHat fails at long distances because UV light is easily absorbed by the atmosphere, limiting its range. However, FR systems are typically performed at close distances, which mitigates this drawback.

## 6.3. UV Safety.

All of our physical-world experiments are conducted in controlled indoor environments. All participants are trained and fully informed about the experimental procedures. To mitigate potential risks, all volunteers are equipped with protective goggles to prevent any exposure to UV light.

# 7. Conclusion

We propose UVHat, a novel and invisible physical attack vector capable of executing multiple adversarial attacks on FR systems. Unlike previous methods, our approach ensures greater stealth, generates stronger adversarial perturbations to enhance attack effectiveness, and demonstrates robust performance across various viewing angles. To maximize UVHat's efficacy, we simulate UV light sources on curved surfaces and leverage reinforcement learning to optimize attack parameters. Extensive experimental results validate the effectiveness and robustness of our approach.

## Acknowledgements

This work is supported by the National Key R&D Program of China under Grant 2022YFB3103500, the National Natural Science Foundation of China under Grant 62020106013, the Chengdu Science and Technology Program under Grant 2023-XT00-00002-GX, the Fundamental Research Funds for Chinese Central Universities under Grant Y030232063003002.

## Impact Statement

In this paper, we introduce UVHat, an invisible and powerful physical adversarial attack against FR systems that utilizes UV light to generate imperceptible perturbations. This research reveals a novel attack pattern that utilizes invisible UV light and contributes to a broader understanding of security vulnerabilities in FR systems. By highlighting this overlooked threat, UVHat can motivate the development of robust and spectrum-aware defense mechanisms, such as multi-modal sensing and spectral filtering, thereby facilitating the advancement of secure and trustworthy biometric authentication technologies.

However, this work also raises serious security and ethical concerns. The proposed attack may allow malicious actors, including fugitives or other individuals seeking to evade surveillance, to circumvent FR-based identity verification and tracking systems. Such misuse could compromise public safety and undermine trust in security-critical applications of FR technology. We encourage future research to explore effective countermeasures against invisible-spectrum adversarial attacks, including real-time detection of abnormal spectral patterns, the integration of UV/IR filtering in camera hardware, and policy-level responses to regulate the deployment of sensitive spectrum-emitting devices. A comprehensive understanding of both the attack and defense perspectives is essential to ensure the safe and responsible evolution of FR systems.

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

Table 6: Attack success rates at different angles across methods.

| Model / Angle | -15° | 0° | 15° |
|---|---|---|---|
| (Komkov & Petiushko, 2021) | 70% | 74% | 71% |
| (Wang et al., 2024) | – | 92% | – |
| IRHat | 56% | 57% | 54% |
| UVHat | **97%** | **99%** | **96%** |

## A. Comparison with Existing Methods

We provide a detailed comparison between UVHat and existing methods, along with experimental results.

**Qualitative comparison.** First, compared to sticker-based methods, our UV light is invisible to the naked eye, providing stronger concealment. As shown in Figure 1, manufacturers deliberately incorporate a small amount of visible light to signal that the UV lamp is active. This design is primarily for safety reasons, as prolonged exposure to UV radiation can cause significant harm to human skin, potentially leading to burns or even an increased risk of skin cancer. Additionally, attackers can freely choose the timing of the attack, offering greater flexibility. Second, compared to visible light-based methods, our invisible light offers superior concealment. Finally, we provide a detailed comparison with infrared-based methods. According to the photon energy formula in physics:

$$E = \frac{hc}{\lambda} \quad (23)$$

where $E$ is the photon energy, $h$ is Planck's constant, $c$ is the speed of light, and $\lambda$ is the wavelength of light. Since UV (10nm-400nm) has a shorter wavelength than IR (700nm-1000nm), it causes stronger adversarial perturbation. Figure 1 confirms that UVHat induces greater disruptions than IR, leading to a higher ASR. Furthermore, existing IR-based attack methods either pose risks to the attacker's eyes or only succeed from a single angle. In contrast, our approach is harmless to the attacker and provides greater flexibility by effectively attacking from multiple angles.

**Quantitative comparison.** To further demonstrate the effectiveness and novelty of UVHat, we compare it with related works. As shown in Table 6, we reproduce the patch-based work (Komkov & Petiushko, 2021) and refer to results in (Wang et al., 2024) (due to lack of devices). Additionally, IRHat replaces UV emitters with IR emitters. As can be seen from the table, UVHat outperforms all methods across multiple angles. Specifically, even though UVHat was placed on a hat, it outperformed the scheme in (Wang et al., 2024) because UV creates greater interference. IRHat's ASR is lower than (Wang et al., 2024)'s ASR, possibly because the glass is positioned closer to the center of the face, making the interference more effective. Note that the UVHat's ASR

is much greater than IRHat's ASR, which further confirms that UV produces more interference than IR.

## B. How the Probability $p$ is Derived?

We define the reward function in Equation 12. Different models compute euclidean distance or cosine similarity for embedding features. To improve clarity, we express FR results as probability $p$ and explain how the probability $p$ is derived from euclidean distance or cosine similarity. First, for euclidean distance $d$, we transform distances using reciprocal and apply softmax:

$$p_i = \frac{e^{1/d_i}}{\sum_{j=1}^{N} e^{1/d_j}}, i = 1, ..., N \quad (24)$$

The smallest distance $d_i$ corresponds to the highest probability $p_i$. The probability threshold $p_\tau$ is derived as:

$$p_\tau = \frac{e^{1/d_i}}{e^{1/d_i} + (N-1)e^{1/d_{avg}}} \quad (25)$$

where the average distance $d_{avg}$ for all non-matching pairs is:

$$d_{avg} = \frac{1}{N-1} \sum_{(x_i,x_j) \in non-match} d(x_i, x_j) \quad (26)$$

where $d(x_i, x_j)$ denotes the $d$ between different embedding features. For a face to be classified as the $i$-th person, its probability must satisfy $p_i > p_\tau$.

Second, for cosine similarity $s$, we apply softmax directly, i.e., $p = softmax(s)$. The highest similarity corresponds to the highest probability, following a process similar to euclidean distance.

