# OpenReview forum: "Omni-Angle Assault: An Invisible and Powerful Physical Adversarial Attack on Face Recognition"
_ICML.cc/2025/Conference — ICML 2025 poster_

### Official Review · Reviewer_ypUC · 2025-02-26

**Overall Recommendation:** 4

**Summary:**

This work introduces UVHat, a physical adversarial attack against face recognition systems that leverages invisible ultraviolet light emitted from a hat. The proposed approach overcomes the limitations of previous methods, particularly regarding visibility and robustness. It is effective in black-box settings and maintains efficacy from multiple angles, rendering detection and mitigation challenging. Extensive experiments conducted in controlled environments demonstrate high attack success rates across various face recognition models.

Shortcomings:
1. The innovative aspects of this approach have not been fully elucidated and require further clarification and justification to highlight their uniqueness.
2. The design of the proposed technical modules needs additional detail to more comprehensively illustrate the underlying principles.
3. The explanation in the experimental section is insufficient and urgently requires more detailed data support and clarification.

**Claims And Evidence:**

Yes, the claims are clearly supported in the paper.

**Essential References Not Discussed:**

Related works are adequately discussed.

**Experimental Designs Or Analyses:**

Yes, the experimental designs and analyses effectively support the proposed methods.

**Methods And Evaluation Criteria:**

Yes, the proposed methods and evaluation criteria make sense for the problem.

**Other Comments Or Suggestions:**

It would be more informative for the authors to further discuss how your findings inspire the development of  adversarial attack on large language models based face recognition.

**Other Strengths And Weaknesses:**

No.

**Questions For Authors:**

1. The current work also uses invisible light to disrupt facial information. In contrast, the proposed mechanism employs a different emitter carrier—namely, a hat. Beyond this, what are the most significant differences or advantages in terms of scenarios and technology? The authors should provide further explanation to clarify the contributions regarding innovation.
2. Regarding the technical modules of the proposed mechanism, I have several questions that require further clarification: Concerning the ultraviolet light source module, the paper mentions that a stealth attack is achieved by regulating the emission angle and intensity. Specifically, how is this regulation mechanism designed? In terms of the emitter carrier design, particularly with the use of a hat as the carrier, what key hardware parameters (such as size, weight, power supply, etc.) need to be met? How are these parameters optimized to ensure efficient invisible light emission while maintaining user comfort?
3. In the experimental section, there are a few points that need further explanation or discussion: In the experimental setup, were multiple lighting conditions (such as daytime, nighttime, artificial lighting, etc.) used to evaluate the system’s adaptability? What impact do these lighting variations have on the performance of the ultraviolet light source module?

**Relation To Broader Scientific Literature:**

1. Compared to visible-light attacks used in existing work, UVHat leverages invisible ultraviolet light to ensure that the attack remains stealthy.
2. The method performs well in black-box attack scenarios, which is a more realistic model of how attacks would work in the real world.
3. The authors compare UVHat against two baseline methods, showing its superior performance in achieving higher attack success rates.

**Theoretical Claims:**

There are no theoretical evaluations in this paper.

---

> ### Author Rebuttal · Authors · 2025-03-30
>
> ***Q-1: What are the most significant differences compared to existing work?***
>
> A-1. Thanks for the comment. We provide a detailed comparison between UVHat and existing methods, along with experimental results.
>
> (1) **Qualitative comparison**
>
> First, compared to sticker-based methods, our UV light is invisible to the naked eye, providing stronger concealment. Additionally, attackers can freely choose the timing of the attack, offering greater **flexibility**.
>
> Second, compared to visible light-based methods, our invisible light offers superior **concealment**.
>
> Finally, we provide a detailed comparison with infrared-based methods. According to the photon energy formula in physics:
> $$
> E=\frac{hc}{\lambda}
> $$
> where $E$ is the photon energy, $h$ is Planck's constant, $c$ is the speed of light, and $\lambda$ is the wavelength of light. Since UV (10nm-400nm) has a shorter wavelength than IR (700nm-1000nm), it causes **stronger adversarial perturbation**. Figure 1 confirms that UVHat induces greater disruptions than IR, leading to a higher ASR. Furthermore, existing IR-based attack methods either pose risks to the attacker's eyes or only succeed from a single angle. In contrast, our approach is **harmless** to the attacker and provides greater flexibility by effectively attacking from **multiple angles**.
>
> (2) **Quantitative comparison**
>
> To further demonstrate the effectiveness and novelty of UVHat, we compare it with related works. The experimental results are summarized below:
>
> |    Model/Angle     | -15° |  0°  | 15°  |
> | :----------------: | :--: | :--: | :--: |
> |       AdvHat       | 70%  | 74%  | 71%  |
> | Wang et al. (2024) |  -   | 92%  |  -   |
> |       IRHat        | 56%  | 57%  | 54%  |
> |       UVHat        | 97%  | 99%  | 96%  |
>
> We reproduce the patch-based work (AdvHat) and refer to Wang (2024)’s results (due to lack of devices). Additionally, IRHat replaces UV emitters with IR emitters. As can be seen from the table, UVHat **outperforms all methods across multiple angles**. Specifically, even though UVHat was placed on a hat, it outperformed the Wang (2024) because UV creates greater interference. IRHat’s ASR is lower than Wang (2024).’s ASR, possibly because the glass is positioned closer to the center of the face, making the interference more effective.  Note that the UVHat’s ASR is much greater than IRHat’s ASR, which further confirms that UV produces more interference than IR.
>
> ***Q-2: How is this regulation mechanism designed? What key hardware parameters need to be met? How are these parameters optimized?***
>
> A-2. Thanks for the comment.
>
> (1) Using the method described in Section 4.1, we determine the UV image based on wavelength, power, and distance. Additionally, the position of UV emitters directly impact the power. Therefore, it is necessary to adjust parameters based on the emitter's position on the curved surface to simulate real-world UV light sources. Specifically, the regulation mechanism utilizes a 3D hemispherical model to analyze the relationship between different positions and UV intensity, then calculates the corresponding UV power accordingly. With the regulation mechanism described in Section 4.2, UVHat can effectively simulate real-world UV light in optimization processes based on attack parameters such as distance, power, wavelength, and position.
>
> (2) We have provided the detailed dimensions and parameters of the attack equipment in Appendix A.
>
> (3) In our optimization, we use Critic-Actor networks to implement reinforcement learning (RL) and find optimal attack parameters for different objectives. The Critic evaluates the current state or state-action pair, and the Actor optimizes the policy using the Critic’s feedback to select the best action. Both networks rely on a reward function linked to model’s predictions and attack goals.
>
> The RL process:
>
> - Initialize the network parameters and value functions.
>
> - At each time step $t$:
>   - Actor selects an action.
>   - Action is executed, obtaining reward & next state.
>   - Critic updates value function.
>   - Actor updates policy via Critic’s feedback.
>
> - Repeat until the process (e.g., max steps).
>
> ***Q-3: Were multiple lighting conditions (such as daytime, nighttime, artificial lighting, etc.) used to evaluate the system’s adaptability? What impact do these lighting variations have on the performance of the UV light source?***
>
> A-3. Thanks for the comment. Yes, we evaluated the robustness of our approach under different ambient light intensities. **Table 5** in our paper presents the attack success rates (ASR) of our method across various lighting conditions. As the ambient light intensity increases, the ASR decreases. This is because higher ambient light levels reduce the visibility of UV light. In most real-life cases, most of the face recognition systems are deployed on indoor scenarios, where the light intensity is usually below 1000 lux, whereas the UVHat attack still shows high effectiveness within these impact factors.

---

> > ### Comment · Reviewer_ypUC · 2025-04-03
> >
> > Thanks for the responses. I have read the author's responses as well as comments from other reviewers. The authors have provided more results and discussions regarding my concerns (the method details, insight of algorithms, and more experimental results etc.).

---

### Official Review · Reviewer_sMwP · 2025-03-07

**Overall Recommendation:** 1

**Summary:**

This paper presents a physical adversarial attack utilizing UV light to disrupt the decision-making of face recognition models. The methodology encompasses physical testing simulations, the implementation of UV emitters, and a reinforcement learning algorithm to optimize attack parameters. While the approach is evaluated on a sufficiently diverse set of datasets and models, the four attack scenarios lack clarity.
## Update after Rebuttal
I regret to say that after two rounds of rebuttal, I still have critical concerns regarding the motivation of this work and do not see any machine-learning-driven insights. The authors primarily highlight security concerns by introducing a new spectrum of light.

Q1: MaxUV achieves a maximum 66% ASR in DoS attacks, while the ASR for targeted impersonation attacks is only 4%. **These results clearly demonstrate that the effectiveness of the untargeted attack is largely due to the significant distortion caused by light covering the face in camera-captured images,** although the proposed method improves ASRs by 23% (actual performance). To demonstrate that the proposed method is an effective machine learning algorithm, the authors must eliminate the impact of algorithm-non-specific distortions, which is challenging in designing untargeted attacks. This is why I suggest focusing solely on targeted attack evaluation.

Q2: The rebuttal states: "The targeted attack is considered successful if the attacker is classified as any one of these 10 target identities." However, **this definition of a targeted attack in adversarial machine learning is incorrect.** A targeted attack should aim for a single, specific identity, allowing the machine learning algorithm to optimize towards a well-defined objective. Furthermore, this definition is particularly inappropriate in face recognition, as human faces share common features. Some prior works refer to this as the "universal face" or "average face".

Q3: I observe inconsistencies between this response and the formulation provided in the rebuttal. It seems that the objective function remains the same, but the only difference is whether the face recognition models are trained with or without the attacker's identity. The formulations show:
* If the untargeted attack is **early-stopped** and the target model is trained **with** the attacker's identity, it is classified as a dodging attack.
* If the untargeted attack is **early-stopped** and the target model is trained **without** the attacker's identity, it is classified as an untargeted attack.
* If the untargeted attack maximizes cross-entropy **without early stop**, it is a DoS attack.

None of these are different from general adversarial machine learning studies.

Q4: This explanation is satisfactory. However, the inputs to the face recognition model are still images with the applied light? Please note that robustness of machine learning refers to the model. If the input is an image with light, it is not surprising that the model makes errors.

Q5: Based on the rebuttal, the difference between this work and Wang et al. (2024) is only in the light spectrum and performance improvement. I expect a machine-learning-driven gap analysis between methodologies, especially from a mathematical perspective.

Q6: The response still focuses on explaining the function of the selected technique, which I refer to as describing "what" this component does, rather than the fundamental reasoning behind selecting it. Similar to Q5, the authors are expected to discuss the theoretical gaps between different technique selections. This is the type of contribution expected in an ICML paper.

Q7: There was no benchmark comparison in the initial submission (which was later included in the rebuttal), and the SOTA method could not be compared even during rebuttal. A fair benchmark comparison must be conducted under the same conditions, particularly for physical attacks (too many variances), to ensure fairness. Simply citing numbers from another paper does not provide a convincing argument.

Q8: I cannot evaluate this without the training settings, but I realize I forgot to request them in my rebuttal comments. Additionally, the concern is similar to Q1: the authors need to mitigate the impact of distortion. A minor point, adversarial training (robustness optimization) is designed to classify adversarial examples as their true identity, not just reject the attack like a detection mechanism, so classification accuracy is a more appropriate metric than ASR in this case.

Q9: I am fine with this. It will ultimately be up to the AC to decide whether these revisions can be addressed during camera-ready preparation.

**Claims And Evidence:**

* I am uncertain whether the proposed attack qualifies as an adversarial attack or merely a common corruption attack. To be considered an adversarial attack, the authors must clearly demonstrate that the optimization process can be directed toward specific adversarial objectives. Currently, the attack strategy appears to primarily disrupt the decision-making process of face recognition models using UV light. To address this concern, the authors should explicitly explain how the targeted impersonation attack is formulated and executed.
* Face recognition technologies have been developed to address challenges posed by varying lighting conditions. For example, Apple's Face ID employs a TrueDepth camera with infrared (IR) technology, enabling it to function under diverse lighting environments. The proposed attack should be tested against such defenses.
* The proposed attack strategy bears similarities to prior attacks leveraging invisible infrared light (e.g., Wang et al., 2024). However, the authors neither discuss the distinctions between their approach and previous work nor provide an empirical comparison.
* 065: The stated limitation of Wang et al. (2024) is unclear. The authors should clarify what is meant by "weak energy" and specify the associated weaknesses in the context of the prior work.

**Essential References Not Discussed:**

No comment.

**Experimental Designs Or Analyses:**

* The paper lacks benchmark comparisons, as the chosen baseline is not strategically selected and remains based on the proposed method. It is unsurprising that an optimized algorithm outperforms its pre-optimization counterpart. While several benchmarked attacks are presented in Figure 1, none are included in the empirical comparisons.
* The evaluation does not consider scenarios where adversarial defenses are present. In particular, adversarial purification techniques could potentially mitigate the proposed attack.

**Methods And Evaluation Criteria:**

* The authors do not clearly explain how the attack incorporates model knowledge, even in the black-box setting, to manipulate decision-making.
* The definition of the impersonation attack, particularly the targeted impersonation scenario, is not well established. It is unclear whether the attack can be directed toward any arbitrarily chosen target, meaning the attack explicitly optimizes toward a specific identity, or if it merely succeeds when the attacker is misclassified as any individual other than themselves. A precise formulation of the attack objective is required.
* **Unclear Attack Scenarios:** The four attack goals lack clarity, particularly DoS and untargeted impersonation. Given that ArcFace and FaceNet are embedding-based face recognition models, it is unclear how these attack objectives are formulated in the feature space. The authors should provide detailed mathematical formulations to clarify how these attack cases are defined and implemented using embedding features.

**Other Comments Or Suggestions:**

No comment.

**Other Strengths And Weaknesses:**

No comment.

**Questions For Authors:**

No comment.

**Relation To Broader Scientific Literature:**

No comment.

**Theoretical Claims:**

The paper lacks in-depth theoretical justification for the proposed methodology. For instance:
* The rationale behind integrating reinforcement learning is not clearly articulated. It remains unclear why reinforcement learning is preferred over a conventional loss function, such as similarity-based optimization.
* Each component of the proposed approach appears to be methodology-driven rather than mathematically grounded.

---

> ### Author Rebuttal · Authors · 2025-03-30
>
> ***Q-1. Provide detailed mathematical formulations to clarify how these attack cases are defined and implemented using embedding features.***
>
> A-1. We apologize for possible misunderstandings about UVHat‘s attack objectives. Our adversarial attack on face recognition (FR) targets four FR attack objectives.
>
> For **dodging attacks**, our goal is:
> $$
> f(UVHat(x_a)) \neq y_a, s.t. y_a \in D_{identity}
> $$
> where $f(\cdot)$ represents FR model, $x_a$ denotes the attacker's face with identity $y_a$ (belongs to identity dataset $D_{identity}$).
>
> We define the reward function in Equation(12). Different models compute Euclidean distance or cosine similarity for embedding features.  To improve clarity, we express FR results as probability $p$ and explain **how the probability $p$ is derived from Euclidean distance or Cosine similarity.**
>
> (a) For **Euclidean distance** $d$, we transform distances using reciprocal and apply softmax:
> $$
> p_i = \frac{e^{1/d_i}}{\sum_{j=1}^{N} e^{1/d_j}}, i=1, ..., N
> $$
> The smallest distance $d_i$ corresponds to the highest probability $p_i$. The probability threshold $p_{\tau}$ is derived as:
> $$
> p_{\tau}=\frac{e^{1/{\tau}}}{e^{1/{\tau}}+(N-1)e^{1/d_{avg}}}
> $$
> where the average distance $d_{avg}$ for all non-matching pairs is:
> $$
> d_{avg}=\frac{1}{N-1}\sum_{(x_i, x_j)\in non-match}d(x_i,x_j)
> $$
> where $d(x_i,x_j)$ denotes the $d$ between different embedding features. For a face to be classified as the i-th person, its probability must satisfy $p_i > p_{\tau}$.
>
> (b) For **Cosine similarity** $s$,  we apply softmax directly, i.e., $p = softmax(s)$. The highest similarity corresponds to the highest probability, following a process similar to Euclidean distance.
>
> For **DoS attacks**, the goal is:
> $$
> f(UVHat(x_a))=None
> $$
> The reward function maximizes entropy:
> $$
> R_{DoS}=-\sum^N_{i=1}p_i log(p_i)
> $$
> This formulation forces the FR model into extreme uncertainty, rejecting recognition queries.
>
> For **untargeted impersonation attacks**, the goal is:
> $$
> f(UVHat(x_a)) \neq y_a, s.t. y_a \notin D_{identity}
> $$
> where the attacker’s identity $y_a$ is absent from the database, and misclassification into any identity in $D_{identity}$ is a success.
>
> For **targeted impersonation attacks**, the goal is:
> $$
> f(UVHat(x_a))=y_{target}, s.t. y_a \notin D_{identity}
> $$
> In practice, we consider randomly selecting 10 target identities, meaning that $p_{target}$ has multiple candidates.
>
> If insist, we will provide a detailed explanation for each attack objective and clarify the **relationship between probability $p$ and the model output** in the reward function to prevent any potential misunderstandings.
>
> ***Q-2. Evaluation of defenses such as Apple's Face ID***
>
> A-2. Thanks for the comment. We tested UVHat against Apple's Face ID on an iPhone 13 (limited to one device due to time constraints):
>
> | Method/Angle                                             | -10°  | 0°    | 10°   |
> | -------------------------------------------------------- | :---: | ----- | ----- |
> | UV light on hat (Successful unlock/Total number)         | 10/50 | 13/50 | 14/50 |
> | Without UV light on hat (Successful unlock/Total number) | 50/50 | 50/50 | 49/50 |
>
> Results show UV light can disrupt Face ID, highlighting the need for improved defenses in IR+RGB systems. Many companies (e.g., Amazon, Mastercard, Hikvision) still use RGB-based FR, making UVHat a serious threat. We also tested adversarial training on FaceNet:
>
> | Percentage of adversarial examples | 5%   | 10%  | 20%  |
> | ---------------------------------- | ---- | ---- | ---- |
> | ASR of UVHat                       | 97%  | 93%  | 90%  |
>
> Adversarial training has limited effectiveness since UVHat optimizes multiple attack parameters (e.g., position, power, number of UV emitters). If insist, we will add more defense tests.
>
> ***Q-3. Discuss the distinctions between UVHat and previous works and provide empirical comparisons.***
>
> A-3.  Thanks for the comment. Please refer to **Reviewer ypUC’s Q-1**.
>
> ***Q-4. Explain how the attack incorporates model knowledge and the rationale behind integrating reinforcement learning.***
>
> A-4. Thanks for the comment. We use Critic-Actor RL to optimize attack parameters for different objectives: As detailed in **Answer 1**, the **reward function is related to model's predictions** and is optimized based on different attack objectives. We conducted experiments with other heuristic algorithms to validate the effectiveness of RL. For details, please refer to **Reviewer B3e4's Q-2 and ypUC's Q-2**.
>
> ***Q-5. The UVHat appears to be methodology-driven rather than mathematically grounded.***
>
> A-5. Thanks for the comment. Section 4.2 provides a detailed mathematical analysis of UV intensity across different positions. Under the black-box setting, we employ RL to optimize attack parameters since white-box methods (e.g., gradient descent) are not applicable. Finally, experiments confirm UVHat’s effectiveness. For details, please refer to **Reviewer B3e4's Q-2**.

---

> > ### Comment · Reviewer_sMwP · 2025-04-02
> >
> > About Motivation
> >
> > 1. The rebuttal still does not address my argument that the proposed attack is fundamentally a common corruption attack. Based on Table 2, increasing Voltage consistently raises ASRs for untargeted attacks, but excessive Voltage only reduces the success of targeted attacks. This may be due to the increased illumination covering more of the face region, as shown in Figure 1, which inevitably decreases recognition accuracy. Given this, the primary evaluation should focus on targeted attacks rather than untargeted ones (such as DoS), as the proposed untargetd attack effectively functions like partially covering the face with a mask.
> >
> > 2. The statement in the rebuttal "In practice, we consider randomly selecting 10 target identities" is unclear. Does this mean that the target is chosen from any of the ten identities, or that the targeted attack is conducted only on these ten specific identities?
> >
> > 3. The rebuttal provides formulations for four attack objectives, but these are difficult to follow for non-expert readers. The authors should refine these descriptions in the introduction and Section 3. Furthermore, what is the distinction between dodging attacks and untargeted impersonation? Why does it matter whether the attacker is included in the identity dataset?
> >
> > 4. I agree with Reviewer B3e4 that this attack may not be truly invisible. As shown in Figure 1, can humans perceive the light during the attack? If not, why does the camera capture the scene differently? This may be a crucial foundation of the attack.
> >
> > About Novelty
> >
> > 5. Multiple reviewers have raised the same concerns regarding novelty. The rebuttal does not establish a fundamental theoretical difference between the proposed attack and existing approaches; it only highlights superior performance and differences in implementation. The objective formulations provided are essentially the same as those used in standard adversarial attacks.
> >
> > 6. Multiple reviewers noted the lack of sufficient theoretical foundation in the paper. However, the rebuttal does not address this issue, focusing only on "how" the attack is performed rather than explaining "why" using these techniques.
> >
> > About Performance
> >
> > 7. A comprehensive comparison with Wang et al. (2024) is necessary, as it represents the latest and most relevant work. Such comparisons, including with other benchmarks, should have been conducted before submission rather than being deferred to the rebuttal stage. It is not a valid excuse that the authors cannot implement Wang et al. (2024) at this stage.
> >
> > 8. The evaluation on FaceID is promising, but the adversarial defense results are not. The adversarial training was conducted with only 20% adversarial examples, which is far below standard adversarial training settings (refer to works in RobustBench). Additionally, how does the attack perform against purification-based defenses, such as DiffPure?
> >
> > About Revision
> >
> > 9. Based on all rebuttals, the paper requires significant revision. How do the authors plan to accomplish this within the 8-page limit? Please note that the appendix is intended for mathematical proofs and less critical content, not for extending the paper length.

---

> > > ### Author Response · Authors · 2025-04-03
> > >
> > > ***Q-1. The attack is a common corruption attack. The evaluation should focus on targeted attacks.***
> > >
> > > A-1.Common corruption attacks occur naturally (e.g., rainy), and are not optimized for models. In contrast, our UVHat introduces carefully crafted perturbations and is optimized for different goals. Clearly, our approach is an adversarial attack.
> > >
> > > We believe that the evaluation should not be biased towards any attack.
> > >
> > > (a) DoS: Higher voltage expands UV coverage to evade FR detection, but not for other attacks.
> > >
> > > (b)Dodging & Untargeted: UV emitter placement affects perturbations and coverage, e.g., a 4V dodging attack may cover less area than a 3V one.
> > >
> > > (c) Targeted: Above 3V, perturbations become too strong, and adjusting emitter placement still fails to classify the attacker as the target identity.
> > >
> > > ***Q-2. “10 target identities" is unclear.***
> > >
> > > A-2. The Targeted attack is considered successful if the attacker is classified as any one of these 10 target identities. Note that Wang et al. found that 15 volunteers could only mimic 364 identities, and subsequently randomly selected 5 targets from these 364 candidate targets. Essentially, Wang et al.'s targets were all identities that the volunteers were likely to attack successfully. In contrast, our approach of randomly selecting 10 targets better aligns with real-world attack scenarios. For instance, an attacker may specifically aim to be recognized as one of 10 company executives.
> > >
> > > ***Q-3. Why does it matter if the attacker is in the identity dataset?***
> > >
> > > A-3. We present real-world examples for non-expert readers.
> > >
> > > (a) Dos: A fugitive evades FR tracking.
> > >
> > > (b) Dodging: FR systems in public places (e.g., government buildings) often use blacklists. An attacker on the blacklist is misidentified as someone off the list, allowing unauthorized access.
> > >
> > > (c) Untargeted: An outsider impersonates an employee to enter a company.
> > >
> > > (d) Targeted: An attacker fools FR to assume a specific identity (e.g., unlocking a phone, gaining access).
> > >
> > > ***Q-4.Why does the camera capture the scene differently?***
> > >
> > > A-4. Humans cannot perceive light during our attack. The different scenes captured by the camera is because its broader wavelength range compared to human eyes (only perceive visible light). Since UV light falls outside the visible spectrum, our method demonstrates strong concealment.
> > >
> > > ***Q-5. The concerns regarding novelty.***
> > >
> > > A-5. We apologize for possible misunderstandings about the novelty. We are the first to reveal that UV lights can disturb cameras and pose security threats to FR systems. Our response to Reviewer ypUC's Q-1 details the differences and advantages over existing works through qualitative and quantitative comparisons. The superior experimental performance further validates the effectiveness of UVHat. Apple’s Face ID experiments further confirm the potential real-world threat posed by UVHat.
> > >
> > > ***Q-6. Explain "why" using these techniques***
> > >
> > > A-6. We explain the reasons and challenges in Section 4.1-4.3.
> > >
> > > (a) Section 4.1 is used to simulate UV light under different parameters.
> > >
> > > (b) Section 4.2 is used to determines variations in UV parameters based on position to better simulate the effect of UV light on a curved surface in the physical world.
> > >
> > > (c) The explanation and experimental comparison about Section 4.3 has been answered in Reviewer B3e4’s Q-2.
> > >
> > > ***Q-7. Compare with Wang et al.***
> > >
> > > A-7. We conducted a comprehensive theoretical comparison with Wang et al., including stronger perturbations (energy analysis), and highlighting that their attack works only from a single angle (as Wang et al. acknowledge).
> > >
> > > We can only cite their results for comparison, and experiments show our method achieves a higher ASR than Wang et al. Note that Wang et al. also did not compare their work with any other studies in the experimental section. The reason for this is that the physical world contains numerous variables, and any change in settings can significantly affect the attack results, making such comparisons unconvincing.
> > >
> > > ***Q-8. The adversarial defense results are not promising.***
> > >
> > > A-8. Thanks for the comment. We constructed the same number of adversarial examples (AEs) as clean samples for adversarial training, evaluating UVHat's ASR and the model's ACC on clean data.
> > >
> > > | Percentage of AE | 5%   | 10%  | 20%  | 100% |
> > > | ---------------- | ---- | ---- | ---- | ---- |
> > > | ASR of UVHat     | 97%  | 93%  | 90%  | 81%  |
> > > | ACC of FaceNet   | 99%  | 98%  | 87%  | 73%  |
> > >
> > > The results show that after training with all AEs, the ASR of UVHat decreased to 81%, but the model's accuracy on clean samples dropped to 73%.
> > >
> > > We have already proposed potential defense strategies in our response to Reviewer hfvN. if insist,  we would like to provide purification-based defenses in the following version.
> > >
> > > **Q-9. How do the authors plan to accomplish revision?**
> > >
> > > A-9. We will clarify & adjust to match the page length requirements while include all necessary descriptions.

---

### Official Review · Reviewer_hfVN · 2025-03-11

**Overall Recommendation:** 4

**Summary:**

The paper introduces a novel method for adversarial attacks on face recognition (FR) systems, mounting ultraviolet (UV) emitters on a hat. The paper simulates the characteristics of this novel physical attack, considering the impact of curved surfaces on the light intensity, and proposes optimization techniques to determine the optimal positioning of UV emitters on the hat. The results show that this attack vector may have a practical impact on FR models, posing a threat in the physical world.

**Claims And Evidence:**

Yes.

**Essential References Not Discussed:**

No.

**Experimental Designs Or Analyses:**

Yes.

**Methods And Evaluation Criteria:**

Yes.

**Other Comments Or Suggestions:**

This paper presents a novel attack against FRs. The idea makes sense, and the approach seems both novel and clever in terms of its simplicity and potential effectiveness. The paper is well-structured and easy to understand.
The issue described in the paper is not entirely new, as many previous works have demonstrated how physical attacks can be carried out. Despite that,  the paper shows how a reasoned combination of UV emitters can further push this physical attacks.
The strongest aspect of this attack lies in its methodology and evaluation. The approach dissects the physical attack and provides a set of formulas that will help researchers develop further attack or defense strategies. The evaluation covers experimental results in the physical world, offering various perspectives on the impact of these attacks. I was particularly pleased to see the real-world evaluations, with a detailed discussion of the results.

**Other Strengths And Weaknesses:**

Strengths
- The paper clearly demonstrates that UV-based adversarial attacks against FR systems are effective.
- The real-world evaluation suggests that this attack vector is effective, and the paper provides an insightful discussion of the results.
- The concept of placing UV emitters on a hat for adversarial attacks is novel, cost-effective, and practical.

Weaknesses
- The problem is known, even though the scheme is new.
- The defenses section could be stronger.

**Questions For Authors:**

See comments.

**Relation To Broader Scientific Literature:**

see comments.

**Theoretical Claims:**

Yes.

---

> ### Author Rebuttal · Authors · 2025-03-30
>
> ***Q-1. This paper presents a novel attack against FRs. The idea makes sense, and the approach seems both novel and clever in terms of its simplicity and potential effectiveness. The paper is well-structured and easy to understand. The issue described in the paper is not entirely new, as many previous works have demonstrated how physical attacks can be carried out. Despite that, the paper shows how a reasoned combination of UV emitters can further push this physical attacks. The strongest aspect of this attack lies in its methodology and evaluation. The approach dissects the physical attack and provides a set of formulas that will help researchers develop further attack or defense strategies. The evaluation covers experimental results in the physical world, offering various perspectives on the impact of these attacks. I was particularly pleased to see the real-world evaluations, with a detailed discussion of the results.***
>
> A-1. Thanks for the positive comments!
>
> ***Q-2. The defenses section could be stronger.***
>
> A-2. Thanks for the comment. The main reason for utilizing UV light is that the wavelength range of the RGB camera is larger than the wavelength range of visible light. Without modifying the RGB camera, we propose to introduce infrared (IR) imaging as a defense. Because the UV light (10nm-400nm) is outside the IR spectrum (700nm-1000nm), combining IR and RBG dual-channel imaging helps detect UV interference.
>
> To test whether existing face recognition (FR) systems with IR cameras can resist interference from UV light, we targeted Apple's Face ID. Apple's official website explains about Face ID advanced technology: “The TrueDepth camera captures accurate face data by projecting and analyzing thousands of invisible dots to create a depth map of your face and also captures an **infrared** image of your face. A portion of your device's neural engine — protected within the Secure Enclave — transforms the depth map and infrared image into a mathematical representation and compares that representation to the enrolled facial data.” Therefore, we tested Apple face ID using an iPhone 13:
>
> | Method/Angle                                             | -10°  | 0°    | 10°   |
> | -------------------------------------------------------- | :---: | ----- | ----- |
> | UV light on hat (Successful unlock/Total number)         | 10/50 | 13/50 | 14/50 |
> | Without UV light on hat (Successful unlock/Total number) | 50/50 | 50/50 | 49/50 |
>
> In this experiment, we use a method similar to MaxUV, i.e., we turn the voltage to maximum and place the UV emitter in a position on the hat closest to the camera. We counted the number of successful unlockings with the UV emitter on and the number of successful unlockings with the UV emitter off at different angles. From the above table, we can see that UV light can still effectively interfere with FR systems, even if it applies an IR camera.
>
> Therefore, the combined imaging method with RGB-IR still needs to be further developed to better defend against the interference of UV light. If insist, we will construct models that receive RBG and IR images for FR and improve robustness to UV interference.

---

> > ### Comment · Reviewer_hfVN · 2025-04-02
> >
> > Thanks for your response. I appreciate the test on Apple's Face ID as it further demonstrates the real-world impact of the proposed work.

---

### Official Review · Reviewer_B3e4 · 2025-03-13

**Overall Recommendation:** 3

**Summary:**

This paper presents a novel physical adversarial attack method named UVHat for face recognition (FR) systems. UVHat generates adversarial perturbations by using ultraviolet emitters mounted on a hat. The proposed method mainly consists of three steps: interpolation-based UV simulation, hemispherical UV modeling, and reinforcement learning-based parameter optimization. Experiments conducted on two datasets and four models demonstrate that UVHat enhances the attack success rate and robustness in black-box settings. Additionally, ablation studies are carried out to analyze the relevant factors.

**Claims And Evidence:**

This paper claims to present an invisible adversarial attack in its title. However, as evidenced by the attack effects shown in the paper, the method (ultraviolet light on the hat) is highly detectable, which is in stark contrast to the claim of a covert attack made in the title. Additionally, the paper fails to utilize any metrics of concealment to evaluate the invisibility of the proposed attack.

**Essential References Not Discussed:**

N.A.

**Experimental Designs Or Analyses:**

The experimental designs in this paper can effectively evaluate the effectiveness of the proposed attack, but there is a lack of evaluation regarding its concealment.

**Methods And Evaluation Criteria:**

This paper measures the attack effectiveness of the proposed attack both in the digital world and the physical world. However, it lacks the measurement of the concealment.

**Other Comments Or Suggestions:**

See Weakness

**Other Strengths And Weaknesses:**

Strengths:

1、The use of ultraviolet light for physical adversarial attacks in this paper is a novel concept. It provides a new direction for the field of adversarial attacks on face recognition systems.

2、This paper is easy to follow.


Weaknesses:

1、The method proposed in this paper mainly lies in the design of ultraviolet light, which seems to have no direct connection with machine learning. I'm curious whether the simple direct use of ultraviolet light can still achieve the attack performance demonstrated in this paper. The current version of this paper seems to me more like an experimental report on physical attacks using ultraviolet light, without reflecting the breakthroughs in machine-learning-related algorithms or the technological innovation.

2、The title of this paper emphasizes invisible adversarial attacks. However, in the experimental result figures of the paper, the purple light on the hat is very noticeable, which is clearly in serious conflict with the invisible attacks claimed by the author. Moreover, this paper lacks corresponding invisibility indicators to measure the invisibility of the attacks proposed in the paper. There is no evidence to support the author's claim of invisibility, whether from human visual observation or indicator values.

**Questions For Authors:**

See Weakness

**Relation To Broader Scientific Literature:**

N.A.

**Theoretical Claims:**

This paper is based on an heuristic approach and fails to provide any proofs for the theoretical claims.

---

> ### Author Rebuttal · Authors · 2025-03-30
>
> ***Q-1. The purple light is very noticeable, which is clearly in serious conflict with the invisible attacks. The experimental designs in this paper can effectively evaluate the effectiveness of the proposed attack, but there is a lack of evaluation regarding its concealment.***
>
> A-1. We apologize for possible misunderstandings about the concealment of ultraviolet (UV) light. UV light is **invisible** to human eyes due to its shorter wavelengths (below 400nm), which fall outside the visible spectrum (400nm-700nm). The human eyes cannot detect light in this range because the lens absorbs short-wavelength UV light to protect the retina.
>
> Although some UV devices emit visible blue-purple light for safety, this does not mean UV light is visible. Manufacturers deliberately incorporate a small amount of visible light to signal that the UV lamp is active. This design is primarily for safety reasons, as prolonged exposure to UV radiation can cause significant harm to human skin, potentially leading to burns or even an increased risk of skin cancer. In our experiments, the UV emitters emit invisible UV light, and its operation was controlled precisely. Instead, cameras capture UV light, which is why images are taken with an iPhone 13. **Figure 1** shows the scene as seen by human eyes, where UV light remains invisible, ensuring no visual anomalies. If UV light is close to the visible spectrum, it may appear as a subtle purple, but due to Rayleigh scattering, it is difficult to detect from a distance. Only cameras at close range can detect the interference. In practice, the attack would remain undetected because security personnel typically patrol from a distance.
>
> In our tests, all five volunteers failed to detect UV light during the attacks, confirming its concealment. Strict safety measures were implemented to ensure no harm to participants. If insist, we will clarify the **physical principles** behind UV invisibility and invite more volunteers to further validate the stealth of this attack in real-world scenarios.
>
> ***Q-2: This paper is based on a heuristic approach without theoretical proofs.***
>
> A-2: Thanks for the comment. We chose heuristic algorithms due to the black-box setting, where the attacker cannot access the model's structure or parameters. This also means that white-box mathematical methods, e.g., gradient descent, do not apply. Additionally, since the wavelengths of UV light are discrete, zero-order gradient optimization does not apply to our optimization process. Therefore, heuristic algorithms provide an effective alternative solution.
>
> Common heuristic algorithms include reinforcement learning (RL), genetic algorithms (GA), and particle swarm optimization (PSO). RL is ideal for optimizing long-term goals, while GA and PSO are focused on finding local optima. RL's ability to optimize attack parameters with different reward functions makes it more suited for our problem. To validate RL's effectiveness, we compared its performance with GA and PSO (the ASR of UVHat in untargeted impersonation attacks):
>
> | Optimization | FaceNet | CosFace |
> | :----------: | :-----: | :-----: |
> |    Our RL    |   93%   |   84%   |
> |      GA      |   71%   |   62%   |
> |     PSO      |   64%   |   67%   |
>
> The results above demonstrate that the ASR of RL is much higher than other methods, which proves its advantage. If insist, we will clarify the choice of RL and provide additional experimental comparisons.
>
> ***Q-3: The method has no connection with machine learning. Can the simple direct use of UV light still achieve the attack performance? The paper fails to reflect the breakthroughs in machine-learning-related algorithms or technological innovation.***
>
> A-3.Thanks for the comment. Our work is the first to reveal that UV lights can disturb commercial off-the-shelf cameras and pose security threats to face recognition (FR) systems, where machine learning is used to optimize and validate the attack. Breakthroughs in machine learning algorithms are outside the scope of this study.
>
> The direct use of UV light cannot achieve effective attack results. In the baseline, we design the **MaxUV**, which places the highest-power UV emitter at the closest position to the camera. The experimental results in Table 1 show the performance of MaxUV. MaxUV achieves a maximum ASR of 66% in DoS attacks, while our UVHat achieves 89%, which indicates that our optimization process significantly improves the attack's ASR. Additionally, MaxUV is much lower in other attack goals. For example, the maximum ASR in targeted Impersonation attacks is only 4%, while our UVHat achieves 69%, further validating the superiority of our approach.
>
> Our contribution lies in discovering a new physical attack vector that threatens the security of FR systems, using interpolation-based UV simulation, hemispherical UV modeling, and RL for optimal attack parameters, with experimental validation in real-world scenarios.

---

> > ### Comment · Reviewer_B3e4 · 2025-04-03
> >
> > Thank you for the author's response. However, my two main concerns remain unresolved:
> >
> > 1) The core algorithm of this paper is designed to use UV light to attack the model. Can ordinary, non-algorithm-driven UV light (such as the common UV lights found in shopping malls) successfully attack the model?
> >
> > 2) Since the author mentions that UV light is invisible to the naked eye, how was the device used to capture images? For instance, can photos taken with a phone or camera retain the UV light that is otherwise invisible to the human eye?

---

> > > ### Author Response · Authors · 2025-04-03
> > >
> > > ***Q-1. The core algorithm of this paper is designed to use UV light to attack the model. Can ordinary, non-algorithm-driven UV light (such as the common UV lights found in shopping malls) successfully attack the model?***
> > >
> > > A-1. Without any algorithm, the ASR of UV light differs significantly from our UVHat, with a maximum gap of 68%. Yes, ordinary, non-algorithm-driven UV light can successfully attack the model, but it needs our pipeline to optimize the attack parameters to achieve better attack results. Notably, the UV emitters used in our experiments are widely available and can be easily purchased in stores. They are commonly used for antique authentication, banknote verification, and fungal detection. In our baseline, MaxUV represents a non-algorithm-driven UV attack. We did not use any simulation or optimization algorithm for MaxUV, we just placed the highest power UV emitter on the hat (placed closest to the camera). The experimental results are shown in Table 1 in our paper, and we selected the experimental results in LFW dataset:
> > >
> > > | Goal       | Method | ArcFace | FaceNet | CosFace | MobileFace |
> > > | ---------- | ------ | ------- | ------- | ------- | ---------- |
> > > | DoS        | MaxUV  | 52%     | 66%     | 41%     | 61%        |
> > > |            | UVHat  | 72%     | 89%     | 80%     | 78%        |
> > > | Dodging    | MaxUV  | 25%     | 32%     | 19%     | 21%        |
> > > |            | UVHat  | 81%     | 100%    | 78%     | 87%        |
> > > | Untargeted | MaxUV  | 33%     | 41%     | 36%     | 28%        |
> > > |            | UVHat  | 77%     | 93%     | 84%     | 80%        |
> > > | Targeted   | MaxUV  | 3%      | 2%      | 0%      | 4%         |
> > > |            | UVHat  | 46%     | 69%     | 44%     | 55%        |
> > >
> > > The results show that MaxUV can successfully attack FR models, while our UVHat can significantly improves the ASR. Specifically, MaxUV achieves a maximum ASR of 66% in DoS attacks, whereas UVHat reaches 89% under the same conditions. Notably, in Targeted impersonation attacks against FaceNet, MaxUV achieves only a 2% ASR, while UVHat reaches 69%, greatly enhancing the effectiveness of UV-based attacks. Therefore, the experimental results validate our claim that our approach significantly improves the success rate of UV light attacks on FR models.
> > >
> > >
> > >
> > > ***Q-2. Since the author mentions that UV light is invisible to the naked eye, how was the device used to capture images? For instance, can photos taken with a phone or camera retain the UV light that is otherwise invisible to the human eye?***
> > >
> > > A-2. Most cameras in daily life can capture UV light. In our experiments, we used the rear camera of an iPhone 13 to capture images, with the device shown in Appendix Figure 7.
> > >
> > > Yes, a phone or camera can capture UV light that is invisible to the human eye. This is because the wavelength range of UV light fall within the wavelength range captured by the camera but outside the visible spectrum for the naked eye.
> > >
> > > In everyday life, this can be observed with UV lamps in malls—cameras capture more bluish-purple light compared to the image seen by the naked eye. Note that the bluish-purple light is intentionally added by manufacturers to indicate the lamp is active (safety reasons), and it does not mean that the UV light is visible to the naked eye. Similarly, infrared light is also invisible to humans. A simple example is pointing a remote control at a smartphone camera while pressing a button—you can see a red dot on the phone's screen, but not with the naked eye.

---

### Decision · Program_Chairs · 2025-05-01

**Decision:**

Accept (poster)

**Comment:**

The submission received one reject, two accepts, and one weak accept. The authors provided detailed responses, which, while not overturning the negative rating, convinced the Area Chair that the submission's strengths and contributions outweigh its weaknesses.